# Analysis of the Effect of Tumor-Grade Change on the Prognosis of Retroperitoneal Sarcoma

**DOI:** 10.3390/cancers14123020

**Published:** 2022-06-19

**Authors:** Sung Jun Jo, Kyeong Deok Kim, So Hee Lim, Jinseob Kim, Min Jung Kim, Jae Berm Park, Kyo Won Lee

**Affiliations:** 1Department of Surgery, Samsung Medical Center, Sungkyunkwan University School of Medicine, Seoul 06351, Korea; sungjunn.jo@samsung.com (S.J.J.); skymomo92@naver.com (S.H.L.); jaeberm.park@samsung.com (J.B.P.); 2Department of Surgery, Inha University Hospital, Inha University School of Medicine, Incheon 22332, Korea; kdkim0609@inhauh.com; 3Department of Epidemiology, School of Public Health, Seoul National University, Seoul 08826, Korea; kimjinseob@snu.ac.kr; 4Department of Surgery, Seoul Medical Center, Seoul 02053, Korea; mjkim@seoulmc.or.kr

**Keywords:** grade change, RPS, second LR

## Abstract

**Simple Summary:**

Changes in tumor differentiation have been observed during local recurrence in retroperitoneal sarcoma (RPS), and these changes have been reported to affect prognosis. However, the change in the tumor grade from the primary tumor to the first local recurrence, and the effect of this change on long-term outcomes, are unknown. This study reports the presence of tumor-grade changes and their effect on patient prognosis. While the grade changes did not affect the patient’s prognosis, a high grade of the primary tumor was an important factor. In addition, the risk factor for second local recurrence was a high grade of the recurrent tumor. Although this finding cannot change the treatment plan for recurrent RPS, it can provide the details of nomograms to predict the patient prognosis.

**Abstract:**

In retroperitoneal sarcoma (RPS), the change in the tumor grade from the primary tumor to the first local recurrence, and the effect of this change on prognosis, are unknown. The aim of this study is to analyze whether these changes affect the prognosis of RPS. Patients who underwent surgery for a first locally recurrent RPS at Samsung Medical Center from January 2001 to February 2020 were included. The pathologic features of primary and recurrent tumors were compared, and the outcomes were measured. A total of 49 patients were investigated. There were 25 patients with different grades of primary and recurrent tumors. The improving, stable, and worsening groups contained 16 (32.7%), 24 (49%), and 9 (18.3%) patients, respectively. There was no significant difference in the prognosis between the three groups. In the analyses of the factors that affect the OS, a high grade of the primary tumor (*p* = 0.023) and the size of the recurrent tumor (*p* = 0.032) were statistically significant in both univariate and multivariate analyses. In a factor analysis of the second LR, a high-grade recurrent tumor (*p =* 0.032) was the only significant factor. There were tumor-grade changes between the primary tumor and recurrent tumor in RPS. However, the most important factor in prognosis is a high grade of the primary tumor.

## 1. Introduction

Retroperitoneal sarcomas (RPS) are rare cancers that are derived from the mesenchymal origin of the retroperitoneum. The main treatment for RPS is surgical resection. Extensive resection that included the peripheral organs as the surgical margin significantly improved the overall survival (OS) [1,2,3]. Even after such extensive resection, RPS is still a difficult disease to treat due to its high local recurrence (LR) (5 years: 23.6%-39%; 7 years: 37.4%) [4,5,6,7]. 

The most important thing in the treatment of relapsed RPS patients is surgical resection. Patients who underwent surgery for recurrence had better prognoses than nonsurgical patients (2 years: 62.8% vs. 26.4%, respectively; 3-year OS: 26.7% vs. 20.8, respectively; 5 years: 43% vs. 11%, respectively) [8]. In addition, there are several studies on perioperative treatment to reduce recurrence after surgery [9,10,11,12].

Interestingly, in recurrent tumors, changes in the tumor differentiation are sometimes found [13,14]. C. Nessim et al. reported on the effects and characteristics of these transformations on prognosis [15]. However, due to the nature of this study as a multicenter study, a centralized pathologic review was not possible. Therefore, the effect of the changes in the tumor grade on the patient prognosis was not included. In addition, there is a limitation in that the tumor grade cannot reflect the characteristics of the entire RPS cohort because the changes in tumor differentiation focus on liposarcoma (LPS).

The purposes of this study were to confirm the existence of changes in the tumor grade and to investigate whether these changes affect the OS and second LR.

## 2. Materials and Methods

### 2.1. Patients

Patients who underwent surgery for a first locally recurrent RPS at Samsung Medical Center from January 2001 to February 2020 were included. Patients were excluded if they: (1) were pediatric patients (under 18 years of age); (2) did not undergo surgery for a recurrent tumor; (3) were diagnosed with visceral sarcoma (tumors that clearly originated from a visceral organ, such as uterine sarcomas and sarcomas of the prostate, bladder, vesicles), extra-skeletal bone sarcoma, dermatofibrosarcoma protuberans, benign tumor, carcinosarcoma, and gastrointestinal tumor. Among the patients who underwent surgery for a recurrent tumor, patients who had undergone palliative surgery were also excluded. 

Data on underlying diseases, gender, BMI, and perioperative treatment and surveillance were investigated through medical records. Tumor histologic subtype, grade, and size were investigated through pathology records. Tumor grade was determined using the French Federation of Cancer Centers Sarcoma Group Grading System (FNCLCC) and was measured in both primary and recurrent tumors [16]. Pathology was performed by dedicated specialized sarcoma pathologists at Samsung Medical Center. Margin assessment (R2 resection) was investigated through operation records. Even with R0/1 resection, according to the operation record, if remnant cancer was found on a CT scan after surgery, it was defined as R2 resection. 

LR was defined as retroperitoneal or intra-abdominal recurrence within the peritoneal cavity, and the duration was calculated based on the date of diagnosis in the follow-up image, not the date of surgery. Second LR was defined as retroperitoneal or intra-abdominal recurrence after surgery for LR, and the duration was calculated based on the date of the first diagnosis in follow-up imaging. 

### 2.2. Definition of Changes in Tumor Grade

The characteristics of the primary tumor and recurrent tumor were compared. In addition, since the entire RPS was targeted, not the LPS, we focused on tumor grade rather than tumor differentiation. We divided patients into three groups (improving, stable, worsening) according to the tumor-grade change. The improving group was defined as having a lower FNCLCC grade of the recurrent tumor than that of the primary tumor. Conversely, if the FNCLCC grade of the recurrent tumor was higher than that of the primary tumor, it was classified under the worsening group. Patients with the same FNCLCC grade for primary and recurrent tumors were defined as the stable group.

### 2.3. Statistical Analysis

Continuous variables with normal distribution are summarized with mean ± standard deviation. For the survival analysis and second local recurrence, Kaplan–Meier estimates and the log-rank test were used. Both the OS and second LR were analyzed using time-to-event regression. The Cox proportional hazard model was used to evaluate prognostic variables, an estimated hazard ratio (HR) with a 95% confidence interval (95% CI) was presented, and *p <* 0.05 was considered statistically significant.

All analyses were performed using R 4.0.4 software (The R Core Team, Vienna, Austria).

### 2.4. Ethical Approval

The study protocol conformed to the ethical guidelines of the Declaration of Helsinki and was approved by the Institutional Review Board of Samsung Medical Center (IRB No. 2021-09-061-001).

### 2.5. Informed Consent

The need for informed consent was waived by the institutional review board of Samsung Medical Center due to the retrospective nature of the study.

## 3. Results

Of the 274 patients who underwent surgery for primary RPS, 95 were eligible according to the inclusion criteria. Patients who did not undergo surgery (*n =* 41) or had palliative surgery (*n =* 3) were excluded. Patients diagnosed with a carcinoma other than sarcoma were also excluded (*n =* 2). The remaining 49 patients were investigated. A flow diagram showing the selection criteria is shown in Figure 1.

### 3.1. Frequency of Changes in Tumor Grade

Among 49 patients, the frequency of the tumor-grade change was 51% (*n =* 25). The group with the greatest tumor-grade change was the improving group (*n =* 16), followed by the worsening group (*n =* 9). In the improving group, most of the patients were 3->2 (*n =* 12), and there was one case of 3->1. In the worsening group, most patients were 1->2 (*n =* 6). In the stable group with no grade change, most patients were 3->3 patients (*n =* 13). The details of the changes are shown in Table 1.

### 3.2. Patient Demographics 

When comparing the three groups according to the tumor-grade change, there was no difference in the clinical characteristics (female-to-male ratio, BMI, underlying disease, perioperative treatment). In the primary tumors, the distribution of the histologic subtypes showed a lower proportion of dedifferentiated liposarcoma (DDLPS) in the worsening group (*p <* 0.001). The distribution of Grade I was also higher in the worsening group at a statistically significant level. In the recurrent tumors, there were no significant differences in the histologic subtypes (*p =* 0.096), tumor grades, tumor sizes (*p =* 0.633), and R2 resections (*p =* 0.726) of the three groups. (Table 2)

### 3.3. Tumor-Grade Change and Prognosis

When the effect of the tumor-grade changes on the OS (*p =* 0.246) and second LR (*p =* 0.543) were analyzed, there was no difference between the three groups (Figure 2). Even when the worsening group was compared to the combined improving and stable groups, there was no significant difference in the OS (*p =* 0.614) and second LR (*p =* 0.38) between the two groups. In the histologic subtype analysis, only the liposarcoma group could be analyzed, and there was no difference between the three groups for both the OS (p = 0.231) and second LR (*p* = 0.549) (Figure 3).

### 3.4. Analyses of Factors Affecting Overall Survival and Second Local Recurrence

In the analyses of the OS, the high grade of the primary tumor and the size of the recurrent tumor were statistically significant in both univariate (Grade III (HR: 2.21, *p =* 0.034), tumor size (HR: 1.01, *p =* 0.007)) and multivariate (Grade III (HR: 2.37, *p =* 0.023), tumor size (HR: 1.01, *p =* 0.004)) analyses. The R2 resection of recurrent tumors was not associated with the OS (*p =* 0.613) (Table 3).

For the analysis of the factors that affect the second LR, 10 patients who underwent R2 resection in recurrent surgery were excluded. In a factor analysis of the second LR, the high grade of the recurrent tumor (HR: 3.32, *p =* 0.032) was the only statistically significant factor. A multivariate analysis was not performed because other factors to be analyzed together could not be found (Table 4).

## 4. Discussion

The current study confirmed that the grade change between the primary tumor and recurrent tumor occurred frequently in RPS, and it showed that this change did not affect the patient prognosis. Moreover, to the best of our knowledge, this is the first study to focus on grade changes rather than on tumor-differentiation changes in LPS.

In retroperitoneal LPS, several studies have reported the existence of the transformation of tumor differentiation, such as the change from well-differentiated liposarcoma (WDLPS) to DDLPS [13,14]. C. Nessim et al. reported how these transformations affect the prognosis. In particular, the change from WDLPS to DDLPS showed the lowest survival rate (6-year OS: 55.6%; 95% CI: 42.7–72.6%) and the highest second recurrence rate (6 years: 76.9%) [15]. In this study, the inconsistency among pathologists caused by not having a centralized review was mentioned to explain why the investigation was conducted by differentiation rather than tumor grade.

On the contrary, we analyzed by tumor grade because a centralized review was possible by way of a single-center study. The hypothesis before the start of the study was that a change in grade within the same differentiation (a change from Grade 2 DDLPS to Grade 3 DDLPS) that is not discriminated only by tumor differentiation would affect the prognosis. However, contrary to the initial hypothesis, the current study showed that changes in the tumor grade did not affect the prognosis. There are various reasons why the change in the grade does not affect the prognosis. One of the factors is that this study investigated all RPS patients, not LPS. By comparing the distribution of the histologic subtypes in each patient group, the worsening group consisted only of LPS, while the improving group included three non-LPS patients. In particular, the histology of the three non-LPS patients in the improving group was malignant peripheral nerve sheath tumor (MPNST) and leiomyosarcoma (LMS), which might have affected the prognosis of this patient group. Another factor to consider is a possible selection bias due to the small number of patients investigated and the inclusion of only patients who had undergone surgery for recurrent tumors. The 5-year OS in this study was 58.4%, which was a little higher than in previous studies (31.6~58%) [7,17,18,19,20]. This result was also influenced by the selection bias that investigated only resectable patients for recurrent RPS. A multicenter study with a centralized pathological review seems to be necessary to clearly understand whether changes in the tumor grade affect the prognosis.

The prognostic factors for recurrent RPS have been reported as a high tumor grade, margin status, and a young age [7,17,20,21,22,23]. Additionally, several factors that may influence the prognosis of RPS have recently been investigated. For example, E. Nizri et al. reported that intraperitoneal invasion was associated with increased LR and decreased survival, and Tyler et al. reported that histologic organ involvement was associated with shorter OS. [24,25]. No additional prognostic factors for RPS were identified in the current study. However, this result seems to be helpful for consideration in situations such as: “Will adjuvant therapy be used when a low-grade primary tumor recurs into a high-grade tumor?”

Previously, there have been several studies on the factors that affect the second LR. On the one hand, Yang JY et al. reported that the grade (HR: 3.32, *p =* 0.045) of the recurrent tumor affects the second LR [21]. On the other hand, Park, J.O. et al. investigated a total of 105 LPS patients and reported that only the growth rate (HR: 2.7, *p <* 0.001) of the tumor affected the second LR [26]. Lewis et al. also mentioned that the presence of the secondary LR complicates complete resection, which affects the prognosis. However, there was no mention of the factors that affect the second LR [27]. The current study showed that a high grade of the recurrent tumor was a prognostic factor for second LR. Although a univariate analysis did not demonstrate factors other than a high grade in recurrent tumors, there is a limitation in that a multivariate analysis was not possible. However, this finding could be used as evidence to analyze the factors that predict prognosis after local recurrence.

For patients with recurrent RPS, complete gross resection might be the only intended treatment option. However, there is no nomogram that can predict prognosis after surgery due to local recurrence, and the risk of complications increases as the operation is repeated [28]. Therefore, the chances of long-term disease-free survival are limited, and this should be recognized in a multidisciplinary evaluation and management plan for individual patients [6,8,29,30]. Although our study does not show results that change the treatment plan for recurrent RPS, it can be used to identify the factors that are used as nomograms to predict prognosis.

The current study has limitations in that it has a retrospective nature, and it investigated a small number of patients. In addition, as mentioned above, there is a selection bias because only patients who underwent surgery for recurrent tumors were selected for the study. The patient distribution also might influence the analysis, as half of the primary tumor grades were concentrated in high grades.

## 5. Conclusions

There were tumor-grade changes between the primary tumor and recurrent tumor in RPS. However, the most important factor in patient prognosis is a high grade of the primary tumor. In addition, a high grade of the recurrent tumor influenced the second LR.

## Figures and Tables

**Figure 1 cancers-14-03020-f001:**
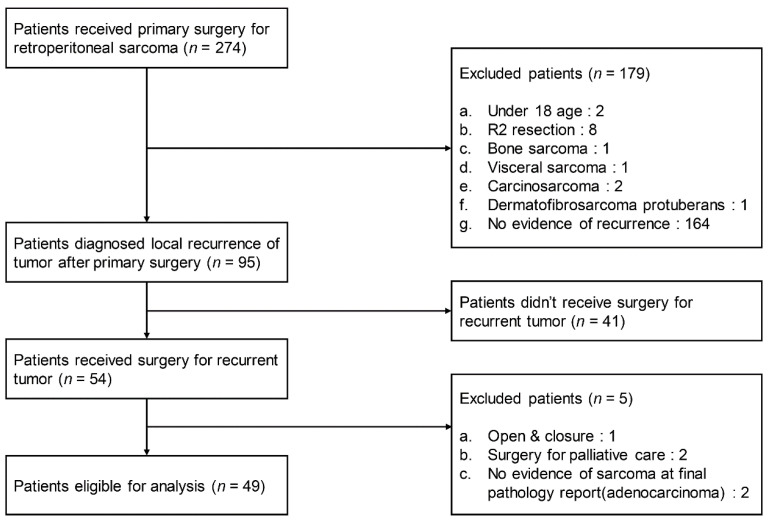
Flow diagram showing the selection criteria.

**Figure 2 cancers-14-03020-f002:**
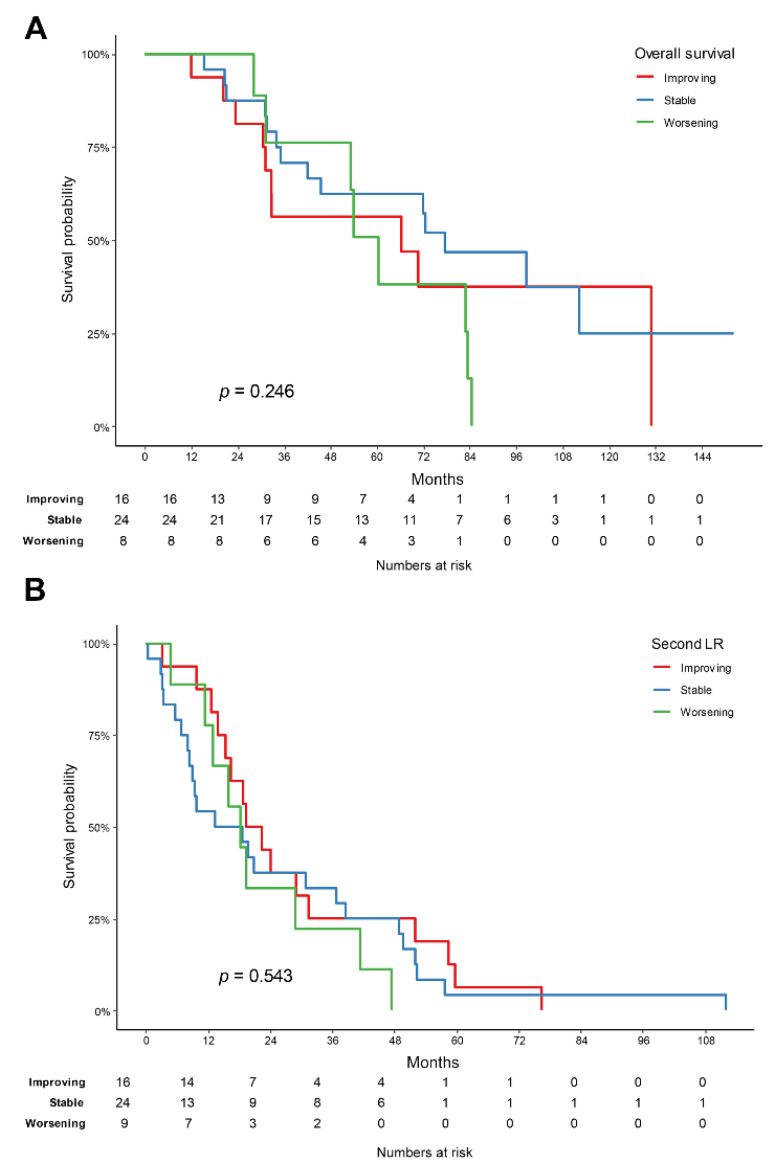
Kaplan–Meier survival graph for each group of improving, stable, and worsening, according to the change in tumor grade: (**A**) overall survival and (**B**) second local recurrence.

**Figure 3 cancers-14-03020-f003:**
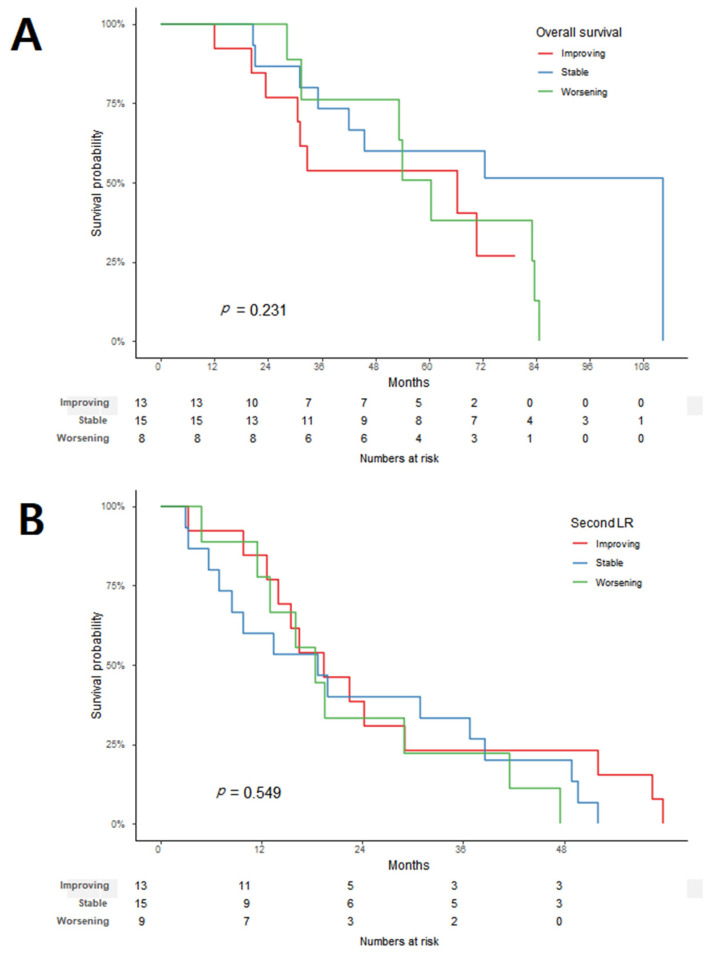
In liposarcoma, Kaplan–Meier survival graph for each group of improving, stable, and worsening according to the change in tumor grade: (**A**) overall survival and (**B**) second local recurrence.

**Table 1 cancers-14-03020-t001:** Distribution of grade changes.

	Primary tumor	Grade 1	Grade 2	Grade 3	n
Recurrent tumor	
Grade 1	2	3	1	*n* = 6
Grade 2	6	9	12	*n* = 27
Grade 3	1	2	13	*n* = 16
n	*n* = 9	*n* = 14	*n* = 26	*n* = 49

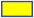
: Improving group 
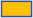
: Stable group 
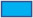
: Worsening group.

**Table 2 cancers-14-03020-t002:** Characteristics of patients.

Variable	Improving Group (*n =* 16)	Stable Group (*n =* 24)	Worsening Group (*n =* 9)	*p* Value
Age (years)	56.77 ± 9.59	50.67 ± 11.11	60.86 ± 8.76	0.033
Sex (*n*, %)				
Male	7 (43.8)	15 (62.5)	6 (66.7)	
Female	9 (56.2)	9 (37.5)	3 (33.3)	0.474
BMI	23.18 ± 3.08	23.12 ± 2.51	23.71 ± 2.22	0.805
Underlying disease (*n*)				
Diabetes mellitus	1	1	1	0.760
Hypertension	6	2	2	0.054
Coronary artery disease	1	0	0	0.184
Primary tumor				
Histologic subtype (*n*, %)				
Well-differentiated liposarcoma	0 (0.0)	2 (8.2)	7 (77.8)	
Dedifferentiated liposarcoma	13 (81.2)	12 (50.0)	2 (22.2)	<0.001
Liposarcoma, NOS	0 (0.0)	1 (4.2)	0 (0.0)	
Leiomyosarcoma	1 (6.2)	1 (4.2)	0 (0.0)	
Malignant peripheral nerve sheath tumor	2 (12.5)	1 (4.2)	0 (0.0)	
Others	0 (0.0)	7 (29.2)	0 (0.0)	
FNCLCC grade (*n*, %)				
I	0 (0.0)	2 (8.3)	7 (77.8)	<0.001
II	3 (18.8)	9 (37.5)	2 (22.2)	
III	13 (81.2)	13 (54.2)	0 (0.0)	
Recurrent tumor				
Histologic subtype (*n*, %)				
Well-differentiated liposarcoma	2 (12.5)	2 (8.3)	0 (0.0)	
Dedifferentiated liposarcoma	11 (68.8)	10 (41.7)	9 (100.0)	0.096
Liposarcoma, NOS (not otherwise specified)	0 (0.0)	3 (12.5)	0 (0.0)	
Leiomyosarcoma	1 (6.2)	1 (4.2)	0 (0.0)	
Malignant peripheral nerve sheath tumor	2 (12.5)	2 (8.3)	0 (0.0)	
Others	0 (0.0)	6 (25.0)	0 (0.0)	
FNCLCC grade (*n*, %)				
I	4 (25.0)	2 (8.3)	0 (0.0)	
II	12 (75.0)	9 (37.5)	6 (66.7)	
III	0 (0.0)	13 (54.2)	3 (33.3)	
Tumor size (mm, mean)	63.12 ± 60.94	61.54 ± 34.73	75.55 ± 37.41	0.633
R2 resection (*n*, %)	3 (18.8)	6 (25.0)	1 (11.1)	0.726
Radiotherapy (*n*, %)				
Given	8 (50.0)	12 (50.0)	5 (55.6)	
Not given	8 (50.0)	12 (50.0)	4 (44.4)	1.000
Chemotherapy (*n*, %)				
Given	1 (6.2)	0 (0.0)	0 (0.0)	0.510
Not given	15 (93.8)	24 (100.0)	9 (100.0)	

**Table 3 cancers-14-03020-t003:** Univariate and multivariate analyses of risk factors associated with overall survival.

Variables	Univariate		Multivariate	
	HR (95% CI)	*p* Value	HR (95% CI)	*p* Value
Improving + stable group vs. worsening group	0.56 (0.25, 1.28)	0.17		
FNCLCC grade of primary tumor				
High grade (III) vs. low to intermediate grades (I+II)	2.21 (1.06, 4.6)	0.034	2.37 (1.13, 4.99)	0.023
Histologic subtype of primary tumor: ref. = DDLPS				
WDLPS	1.14 (0.47, 2.76)	0.765		
LPS, NOS	2.32 (0.3, 18)	0.42		
LMS	0.43 (0.05, 3.46)	0.43		
MPNST	1.43 (0.32, 6.32)	0.637		
Other	0.7 (0.23, 2.19)	0.544		
FNCLCC grade of recurrent tumor				
High grade (III) vs. low to intermediate grades (I+II)	1.74 (0.85, 3.57)	0.131		
Tumor size of recurrent tumor	1.01 (1, 1.02)	0.007	1.01 (1, 1.02)	0.004
R2 resection of recurrent tumor: Yes vs. No	1.79 (0.79, 4.06)	0.163		

**Table 4 cancers-14-03020-t004:** Univariate analysis of risk factors associated with second local recurrence.

Variables	Univariate	
	HR (95% CI)	*p* value
Improving + stable group vs. worsening group	0.71 (0.31, 1.58)	0.398
FNCLCC grade of primary tumor: ref. = 1		
2	0.49 (0.19, 1.25)	0.135
3	0.86 (0.38, 1.94)	0.719
Histologic subtype of primary tumor: ref. = DDLPS		
WDLPS	1.54 (0.68, 3.48)	0.303
LPS, NOS	3.81 (0.47, 30.75)	0.209
LMS	0.27 (0.03, 2.08)	0.208
MPNST	1.66 (0.48, 5.76)	0.422
Other	1.26 (0.42, 3.76)	0.676
FNCLCC grade of recurrent tumor: ref. = 1		
2	2.2 (0.81, 5.93)	0.12
3	3.32 (1.11, 9.94)	0.032
Tumor size of recurrent tumor	1 (0.99, 1.01)	0.615

## Data Availability

The data presented in this study are available upon request from the corresponding authors.

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
