# Peer review of "Analysis of the Effect of Tumor-Grade Change on the Prognosis of Retroperitoneal Sarcoma"

_cancers, 2022, doi:10.3390/cancers14123020_

Round 1

Reviewer 1 Report

In the manuscript by Jo et al. titled, "Analysis of the Effect of Tumor Grade Change on the Prognosis of Retroperitoneal Sarcoma," the authors perform a 20 year retrospective analysis of RPS to see if when the tumor grade changes between primary tumor and local recurrence if the prognosis changes. Despite the fact that this occurs pretty frequently in RPS, the change in tumor grade had no impact on OS. The study was explained properly and the limitations were enumerated in the discussion section so as to assist the reader in determining if there is any merit to the study. I believe the data supports the conclusions. There are only two minor changes to address in my opinion. 

1) There are 5-6 papers in the past 3 years that address the factors that contribute to the prognosis of patients with RPS. A more current literature review on these studies will strengthen the background of the study that these authors have performed. 

2) There is no indication what the +/- represents in the tables. I assume that it is the standard deviation of the mean, but it should be explicitly stated somewhere or multiple areas (i.e. methods section, footnote under the table, or in the results section). 

Author Response

Thank you for your detailed review

1) Papers review on the prognostic factor of RPS and the clinical implication of this study were newly mentioned in the discussion.

2) The meaning of +/- was mentioned in the method part.

Please see the attachment for details.

Reviewer 2 Report

The authors did the good retrospective analyse of different types sarcomes located in the retroperitoneal space. The inclusions criterias and exclusions criterias were described clearly. Unfourtanely the analysed group is unconsitent by cell types. The liposarcoma is moderate on malignancy compare to aggressive types of sarcoma. The conlusions are confused . The size. , type of surgery and recurrence of disease are not the main factors for survivving of patients compre the histological type of sarcoma. This factor is more important. The therapy showed that the irradiationa and chemotherapy has not influence to survive data. That this fact can be true if we analysed the moderaty histological type sarcomas.  This study must be improved on facts of survive influence not only description of the small very selective geoup with no correlation to biological behavvoior of tumor cells types. The absolute survive and five Ears survive of this group werent no compare ti thee more other studies in literature. 

Author Response

Thank you for your detailed review

1) Among cell type, only liposarcoma patients were additionally analyzed for the effect of tumor grade change. In the other cell types, analysis was not possible. We will analyze it in future research.

2) Five year survival data was added and compared with other papers.

Please see the attachment for details

Round 2

Reviewer 2 Report

No comments